# A Strategy for Personalized Treatment of iPS-Retinal Immune Rejections Assessed in Cynomolgus Monkey Models

**DOI:** 10.3390/ijms21093077

**Published:** 2020-04-27

**Authors:** Shota Fujii, Sunao Sugita, Yoko Futatsugi, Masaaki Ishida, Ayaka Edo, Kenichi Makabe, Hiroyuki Kamao, Yuko Iwasaki, Hirokazu Sakaguchi, Yasuhiko Hirami, Yasuo Kurimoto, Masayo Takahashi

**Affiliations:** 1Laboratory for Retinal Regeneration, RIKEN Center for Biosystems Dynamics Research, 2-2-3 Minatojima-minamimachi, Chuo-ku, Kobe 650-0047, Japan; shota.fujii@gmail.com (S.F.); yoko.futatsugi@riken.jp (Y.F.); masaaki.ishida@riken.jp (M.I.); ayaka.edo@riken.jp (A.E.); makabe.k1@gmail.com (K.M.); retinalab@ml.riken.jp (M.T.); 2Department of Ophthalmology, Kawasaki Medical School, 577 Matsushima, Kurashiki, Okayama 701-0114, Japan; hiroyuki_retina_galileogalilei@yahoo.co.jp; 3Department of Ophthalmology & Visual Science, Tokyo Medical and Dental University Graduate School of Medical and Dental Sciences, 1-5-45, Yushima, Bunkyo-Ku, Tokyo 113-8510, Japan; sahnya96@gmail.com; 4Department of Ophthalmology, Osaka University Medical School, 2-2 Yamadaoka, Suita, Osaka 565-0871, Japan; sakaguh@ophthal.med.osaka-u.ac.jp; 5Department of Ophthalmology, Kobe City Eye Hospital, 2-1-8 Minatojima-minamimachi, Chuo-ku, Kobe 650-0047, Japan; yhirami@kcho.jp (Y.H.); ykurimoto@mac.com (Y.K.)

**Keywords:** iPS cells, retinal pigment epithelial cells, immune rejection, drug, transplantation

## Abstract

Recently, we successfully transplanted an autograft, or major histocompatibility complex (MHC)-matched allografts, from induced-pluripotent-stem-cell-derived retinal pigment epithelial (iPSC-RPE) cells in patients with age-related macular degeneration. However, there was an issue regarding immune rejection after transplantation. In this study, we established a preoperational in vitro “drug–lymphocytes–grafts immune reaction (Drug-LGIR)” test to determine the medication for immune rejection using host immunocompetent cells (lymphocytes) and transplant cells (target iPSC-RPE cells) together with different medications. The adequacy of the test was assessed by in vivo transplantation in monkey models together with medication based on in vitro data. In the results of Drug-LGIR tests, some drugs exhibited significant suppression of RPE cell-related allogeneic reactions, while other drugs did not, and the efficacy of each drug differed among the recipient monkeys. Based on the results of Drug-LGIR, we applied cyclosporine A or local steroid (triamcinolone) therapy to two monkeys, and successfully suppressed RPE-related immune rejections with RPE grafts, which survived without any signs of rejection under drug administration. We propose that our new preoperational in vitro Drug-LGIR test, which specifies the most efficacious medication for each recipient, is useful for controlling immune attacks with personalized treatment for each patient after retinal transplantation.

## 1. Introduction

Immunosuppressive medications are used for patients with ocular inflammation in Japan. For instance, patients with uveitis and other ocular inflammatory diseases receive betamethasone and dexamethasone (eye drops), triamcinolone (injection) [1], and prednisolone (oral). For patients with uveitis, immunosuppressants, including biologics such as cyclosporine A and FK506 (oral) [2], infliximab (intravenous) [3], and adalimumab (subcutaneous injection) [4], are used. Although we have proper protocols for the treatment of ocular inflammation diseases, at the moment, we have no answer nor evidence about the treatment of immune rejections after retinal cell transplantation. Compared with pan-uveitis, inflammation around the grafts after retinal cell transplantation might be local, but even a tiny inflammation should be harmful for the host retina and grafted cells after the surgery. 

In September 2014, we successfully transplanted an autograft sheet made from induced-pluripotent-stem-cell-derived retinal pigment epithelial (iPSC-RPE) cells in a patient with age-related macular degeneration (AMD) [5]. As there were no problems noted postoperatively, shown by the survival of the graft RPE sheet and absence of tumors and inflammation, these results confirmed that this novel method of regenerative medicine is promising. However, the disadvantages of autograft transplantation using iPSCs include the fact that it is costly and time consuming [6]. Transplantation requires preparation of stable and high-quality iPSC-RPE cell lines. Therefore, prior to any procedure, it is necessary to decide (1) which cell type is going to be transplanted, (2) how the cells will be transplanted into the subretinal space (surgical procedure), and (3) how the postoperative inflammation will be treated in iPSC-RPE cell transplantation. Particularly, there is no evidence about the latter issue, e.g., regarding which medication should be used after retinal transplantation. The final goal of our study is to establish a new regenerative therapy with iPSC-RPE cells that can be used worldwide, which includes the postoperative medication.

Although the subretinal space is an immune-privileged site [7], RPE allografts usually cause immune responses even in this space. When allogeneic RPE cells were transplanted into the subretinal space of AMD patients [8], there were immune rejection issues. Rejection also occurred when allogeneic RPE cells were transplanted into the subretinal space of rhesus macaques [9]. Furthermore, murine allogeneic RPE cells were rejected early in the postoperative period due to innate immune activity [10]. Human RPE cell xenografts have also been shown to cause inflammation in the subretinal space of experimental animals, e.g., a human fetal RPE cell sheet transplantation into the subretinal space caused severe choroidal inflammation [11] or immunological rejection in rabbits [12]. When human embryonic stem cell-derived RPE cells (ESC-RPE cells; xenografts) were transplanted into the subretinal space of rabbits, these cells were also rejected [13,14,15]. In the present study, we used human iPSC-RPE cells as RPE xenografts that were transplanted into cynomolgus monkeys. The immunogenicity of human iPSC-derivatives varies depending on what type of cells/tissues iPSCs are differentiated into [16,17]. Among these induced cells/tissues, iPSC-RPE cells have been shown to exhibit immunogenicity [18] that can supposedly trigger strong immune responses.

In human RPE allograft transplantation, it will also be important to suppress postoperative inflammation to ensure long-term graft survival. The purpose of this study is to establish a preoperational evaluation system of immunosuppressive agents for the treatment of postoperative immune rejection. In this study, we used the same set of human iPSC-RPE cells (grafts) and normal cynomolgus monkeys (recipients) both in vitro and in vivo. In the in vitro study, we performed drug–lymphocytes–grafts immune reaction (Drug-LGIR) tests using host immunocompetent cells (effector lymphocytes) and transplant cells (target iPSC-RPE cells) with medications. We then performed in vivo transplantation into the monkey models based on the in vitro data and confirmed the efficacy of having a preoperational in vitro drug evaluation system, the Drug-LGIR.

## 2. Results

### 2.1. Results of LGIR In Vitro Tests in Monkeys

First, we performed lymphocytes–grafts immune reaction (LGIR) tests [19,20,21,22], an in vitro system to predict immune rejection prior to transplantation by coculturing host immune cells and target explant cells. In the current study, for host immune cells, we used peripheral blood mononuclear cells (PBMC) from healthy monkeys (HM-1, -2, -3, and -4), and for target cells, we used human iPSC-RPE cells assuming iPSC-RPE transplantation. After 5 days of coculture, the activity of the immune cells against the target RPE cells was evaluated by the proliferation of the immune cells marked by the expression of Ki-67. As a result, representative data showed host immune cells in monkey PBMC responding to target human RPE cells, as well as to allogenic B cells that were used for positive control (Figure 1), although the degree of the reaction varied among individuals (*n* = 4, Appendix A). 

### 2.2. Results of Human iPSC-RPE Xenotransplantation

As the purpose of the LGIR test described above was to assess the immune response prior to transplantation, next we performed xenotransplantation of the human iPSC-RPE cells examined above into six eyes of four monkeys (HM-1 to HM-4) (Appendix A). iPSC-RPE cells were prelabeled with PKH-fluorescent dye that allows tracking of live cells. Since PBMCs from the HM-1 monkey responded well to the human RPE cells in the LGIR test (Figure 1), we first used this monkey for the in vivo transplantation. Similar to the results seen after allogeneic RPE cell transplantation in monkeys [20,21], we found fluorescein angiography (FA) leakages from the transplanted human iPSC-RPE cells, but no abnormal sign around the transplanted area by optical coherence tomography (OCT) in the HM-1 monkey (Appendix A) was observed. Therefore, hematoxylin and eosin (H&E) staining and immunohistochemistry (IHC) evaluations of the retinal sections were also performed to evaluate the grade of immune rejections. As a result, we observed severe inflammation in both eyes, e.g., H&E staining in the right eye showed development of inflammatory granuloma in the subretinal space (Figure 2A, yellow dot-line). In particular, retinal sections from the left eye exhibited inflammatory granuloma along with a large amount of inflammatory cell infiltration in the choroid (Figure 2B, yellow dot-line). To compare the severity between the eyes, we performed IHC with immune cell markers. Although there were some CD3^+^ T cells in the right eye (Figure 2C), there were larger numbers in the left eye (Figure 2D) around the grafted area. However, perhaps more importantly, we were unable to find any live grafted RPE cells that were PKH positive in any of the sections. These results indicated that the grafted RPE cells were not able to survive due to immune rejections.

We also examined whether other types of immune cells invaded the retina and the choroid of the left eye of HM-1. IHC evaluations confirmed that there were various inflammatory cells in these sites (Appendix A), e.g., CD20^+^ B cells were found in the granuloma. In other sections, there were IgG^+^ B cells in the choroid and IgG deposits under the host RPE layer. Unlike the RPE cell allografts in humans [23], xenotransplantation of human iPSC-RPE into monkey HM-1 showed NKG2A^+^ natural killer (NK) cells in the choroid under the grafted RPE cells. Moreover, we also found a large number of interferon gamma (IFN-γ) and CD3 double-positive T cells (perhaps Th1-type helper T cells).

We examined another transplanted monkey (HM-2). The xenotransplantation method was the same as the monkey examined earlier (HM-1). At 4 weeks after the transplantation, cell infiltration was seen in the retina around the explanted site (Appendix A). At 12 weeks, the thickness of the neural retina had decreased over time (Appendix A). Although H&E staining showed the presence of cystoid retinal edema and cell infiltration in the choroid (Appendix A), the inflammation was not as severe as for HM-1. IHC showed a large number of CD3^+^ T cells in the choroid, but PKH^+^ graft RPE cells still existed around the host RPE cell layer (Appendix A). So, in this case, even though there were immune attacks, the graft survived, unlike in the left eye of HM-1. The difference in the survivability of the explant between HM-1 and HM-2 might be due to the difference in the level of inflammation among individuals. Appendix A summarizes the grading of immune rejection in each monkey against the iPSC-RPE cell transplantation.

### 2.3. Results of In Vitro Drug-LGIR Tests

We next applied several drugs, such as dexamethasone, prednisolone, betamethasone, hydrocortisone, triamcinolone, and cyclosporine A, to the LGIR test (Drug-LGIR). Before the assay, we evaluated the best concentration of each drug for the in vitro assay. For this, we assessed the suppression of the activities of monkey lymphocytes in PBMCs under four concentrations of each drug (0.01, 0.1, 1, and 10 μg/mL). After culturing PBMCs in the presence of the drug at each concentration, we evaluated the secretion level of IFN-γ in the supernatants. All drugs greatly suppressed the production of IFN-γ at the concentration around 1 or 10 μg/mL (Appendix A). Therefore, we used the concentration of 1 μg/mL for all drugs in the following experiments. 

In Drug-LGIR assay, PBMCs drawn from transplanted (HM-1 and -2) or nontransplanted (HM-5, -6, -7, and -8) monkeys were cocultured with human iPSC-RPE cells under the presence of the drugs. Then, the in vitro drug response was evaluated in terms of whether the proliferation activities of the immune cells were suppressed by the drugs, e.g., in monkey HM-5, hydrocortisone, prednisolone, and cyclosporine A exhibited significant suppression of the proliferation of the immune cells, assessed by Ki-67 staining of CD4^+^ T cells, CD8^+^ T cells, and CD20^+^ B cells. In contrast, other drugs, such as dexamethasone, betamethasone, and triamcinolone, were not suppressive (Figure 3A). In monkey HM-2 (Figure 3B), dexamethasone exhibited significant suppression of the immune reaction. Prednisolone and triamcinolone also exhibited suppressive effects to some extent. However, other drugs, including cyclosporine A, were not suppressive. The results of Drug-LGIR in six monkeys are summarized in Figure 4A–F. Although the degree of suppression of each drug varied among the recipient monkeys, all drugs had immunosuppressive properties to RPE-related immune responses, except for the case of HM-8 with triamcinolone, which failed to suppress and rather stimulated the proliferation of the immune cells in vitro (Figure 4D). 

### 2.4. Effects of Local Steroid Administration in RPE Cell Transplantation

Based on the results of the Drug-LGIR tests, we next examined whether local steroid administration could control the immune attacks after xenotransplantation. Since triamcinolone was effective in vitro for monkey HM-6 (Figure 4B), we used this medication to control the immune attack. For this experiment, we explanted human iPSC-RPE cells in conjunction with the administration of intravitreal triamcinolone acetonide (IVTA) and sub-Tenon conjunctival injection of triamcinolone acetonide (STTA). We performed the IVTA injection on day 0 (operation day) and the STTA injections at 4 and 12 weeks following transplantation. Evaluation of the RPE cell graft showed no rejection signs around the grafts by fundus (Figure 5A), FA (no leak from the grafts and retina/vessels, Figure 5B), and OCT (Figure 5C) examinations. Histological analysis showed that RPE graft cells survived in the subretinal space (Figure 5D). IHC analysis showed that there were no CD3^+^ T cells (Figure 5E) and only a very small number of Iba1^+^ (Figure 5F) and MHC-II^+^ cells (Figure 5G) throughout the sections. Accordingly, the survival of RPE graft cells (PKH positive) around the host RPE layer was confirmed (Figure 5E–G). Thus, we reasoned that the immune rejection in the eye was able to be controlled under triamcinolone administration in HM-6. 

### 2.5. Effects of Systemic Cyclosporine A Administration in RPE Cell Transplantation

As cyclosporine A was effective in the Drug-LGIR test for monkey HM-5 (Figure 4A), we used this medication for HM-5 to control the immune attack after transplantation. We administrated cyclosporine A before and after the transplantation daily by oral administration with food mixture. The level of cyclosporine A in blood was monitored during the evaluation for 3 months after the surgery. As revealed in Figure 6, there were no rejections around the RPE grafts evaluated by fundus (Figure 6A), FA (Figure 6B), OCT (Figure 6C), H&E staining (Figure 6D), and IHC analysis (Figure 6E,F). By IHC, we found that explanted RPE cell grafts survived in the subretinal space (Figure 6E) and no CD3^+^ T cells around the grafts (Figure 6F). Throughout the sections, we did not find inflammatory cells (Iba1^+^ and MHC-II^+^ cells) in the retina (data not shown). Thus, in HM-5, the immune rejection in the eye was able to be controlled under cyclosporine A administration, which was consistent with the in vitro Drug-LGIR data (Figure 4A). 

### 2.6. Surgical Complications in RPE Cell Xenotransplantation

There were some complications associated with the surgical procedures detected during and after the surgery for RPE cell transplantation, including endophthalmitis and epiretinal membrane (ERM) (Appendix A). As seen in Figure 7, monkey HM-8 developed endophthalmitis 1 week after the transplantation, in which the fundus of the left eye was no longer visible due to inflammation (Figure 7A). However, the fundus findings greatly improved 3 weeks after surgery without any treatment (Figure 7B). Eventually, this inflammation completely disappeared in this case. Importantly, triamcinolone was not effective in vitro for HM-8 in the Drug-LGIR test but was in fact immune stimulative (Figure 4D). Coincidentally, we found triamcinolone-related endophthalmitis in the eye after IVTA injections, which at least did not conflict with the result of Drug-LGIR. 

In three eyes of two monkeys, ERM complications occurred after the transplantation (Appendix A). As seen in the representative case of HM-8, the right eye (Appendix A) color fundus and OCT showed a gradual increase in the ERM thickness after the transplantation, which suggested that intravitreous-grafted RPE cells and inflammatory cells could have generated the ERM. In order to confirm this, we examined the retinal sections by H&E staining and by IHC of grafted RPE cells and inflammatory cells. H&E staining showed pigmented ERM overlying the macula (Appendix A). Interestingly, enhanced green fluorescent protein (eGFP)-positive grafted RPE cells and a large number of inflammatory cells (Iba1^+^, MHC-II^+^, and CD3^+^) were observed in the ERM tissues. These results demonstrated that the ERM was composed of at least transplanted RPE cells and immune cells.

## 3. Discussion

In order to control immune attacks after RPE cell transplantation, we established an in vitro drug assay, termed “drug–lymphocytes–grafts immune reaction (Drug-LGIR)” tests in the current study. Previously, we established a similar test using peripheral blood lymphocytes of the recipient cocultured with iPSC-RPE cells of the explant to predict the immune rejections anticipated after RPE cell transplantation [19,20,21,22]. In the current study, we enhanced the test by applying commonly used medication for ocular inflammatory diseases in Japan. We did an in vitro test followed by in vivo medication administration in four monkeys, confirming that immune rejections in the eyes were able to be controlled under the administration of the medication based on the in vitro data. Besides, we found triamcinolone-related sterile endophthalmitis in the eye after IVTA therapy. This complication was developed only in the monkey that showed immune response to triamcinolone in vitro during the Drug-LGIR test, although generally triamcinolone exhibited a suppressive effect in the Drug-LGIR tests, as shown in the results for other monkeys. This strongly suggests that some drugs may have opposite effects depending on the individuals, and by the Drug-LGIR test, we can predict before transplantation if the recipient has unusual immune responses such as allergies against particular drugs. Since the drug in vitro test lasted for 5 days and was followed by a 1-day experiment of FACS, it takes only a week to obtain the result of the drug prescreening before transplantation.

The present study examined human iPSC-RPE cell transplantation (cell suspension; xenografts) into the subretinal space of cynomolgus monkeys. Most of the cases exhibited intraocular inflammation, which was detected by FA examinations showing fluorescein leakages from the graft sheet and cells and by OCTs showing inflammatory immune cell infiltration or granuloma-like mass lesions at the transplanted sites. IHC examinations revealed invasions of inflammatory immune cells, including T cells and antigen-presenting cells, into the retina. In addition, invasions of IgG-positive B cells and NK cells were observed at the sites of immune rejection in the retina. So, grafted human iPSC-RPE cells were not able to survive due to immune rejections. These results indicate that RPE cells are immunogenic [19,20,21] and, thus, RPE cell transplantation except for autografts might not succeed unless immune rejections are controlled. 

With regard to postoperative treatments after subretinal RPE transplantation, it has been reported that suppression of the host immune responses improves graft survival [24]. In human ESC-RPE subretinal transplantation (allografts), tacrolimus and mycophenolate mofetil have been used for systemic immunosuppression. Rejection was not observed after the use of these drugs, as the graft survived for a long period of time [25]. However, several systemic adverse events related to systemic immunosuppression were observed [25].

Complications associated with systemic immunosuppression are often critical for the host animals [26,27]. On the other hand, local immunosuppression can be achieved by using drugs such as intravitreal cyclosporine A, which has been shown to prolong the graft survival in human fetal RPE xenotransplantation [28]. The necessity and benefit of using either systemic or local immunosuppression probably depends on the type of transplanted cells (iPSC-RPE cells, ESC-RPE cells, or fetal RPE cells) and the form of transplantation (e.g., xenografts, allografts, or autografts).

Individual differences in the response to immunosuppressive drugs is not limited to post-transplantation treatment, e.g., in patients with noninfectious uveitis, the suppression of intraocular inflammation by each drug varies among patients. In the case of uveitis with Vogt–Koyanagi–Harada disease, in one patient, steroid-pulse treatment was highly effective, but this was not the case with other patients. In the case of Behcet’s disease uveitis, in some patients, biologics were remarkably effective, while in other patients no efficacy was shown, and there were persistent inflammatory attacks. In idiopathic uveitis, some patients showed a marked response to STTA therapy, but other patients did not respond and showed repeated recurrence. We assume the possible causes of such difference in drug efficacy as (1) a difference in expression levels of the drug receptors, (2) a difference in individual immune status, (3) a difference in drug transferability among individuals, (4) a difference in individual drug metabolism, (5) the presence of neutralizing (drug resistant) antibodies in the individual, and so on. Given that individual differences in drug response are reasonable, the LGIR tests using host immunocompetent cells (effector lymphocytes in PBMC) and transplant cells (target cells) in vitro may mimic the individual rejection state, and the Drug-LGIR tests using clinical medication may mimic the suppression of the rejection by the drugs in each patient. Based on the results of this preclinical in vitro examination, we might be able to use the exact treatment for each patient to suppress intraocular inflammation leading to rejection after transplantation.

In the present study, the left eyes of monkeys HM-1 and HM-2 after RPE cell suspension transplantation had relatively severe postoperative inflammation, despite the administration of IVTA (Appendix A). This is somewhat confusing, as steroids suppress T cell activation by inhibiting the production of inflammatory cytokines, such as interleukin-2 (IL-2), IFN-γ, and tumor necrosis factor alpha (TNF-α) [29], which indeed inhibit microglial proliferation [30]. Stanzel et al. reported that RPE cells in a single-cell suspension lose their polarized cytoskeletal and membrane-protein distribution, and thus, likely become more susceptible to immune rejection [31]. Therefore, cell suspension transplantations might require higher doses of steroid compared to sheet transplantation [20] to inhibit postoperative inflammation. The amount of IVTA used in this study was half the general human dose, which was possibly lower than it should have been for some of the monkeys. Choudhry et al. have reported that a triamcinolone acetonide injection, whether administered intravitreally or via a posterior sub-Tenon route, is an effective option for the treatment of inflammatory diseases, even though the sub-Tenon route has a lower bioavailability of the drug [32]. In the current study, we applied additional STTA in two monkeys (three eyes) postoperatively, which appeared effective for reducing postoperative inflammation (Figure 5) [21]. Since the STTA treatment combined with IVTA inhibited postoperative inflammation in RPE transplantation, we believe that local steroid therapy by IVTA and STTA will be helpful for successful transplantation. However, IVTA can cause sterile endophthalmitis on rare occasions [33], and it has been reported that it often causes floaters for a period after the injection [34]. As STTA is not accompanied by these complications, it might be better to use STTA as the first option. Further studies on the appropriate usage and dosage of steroids will be required.

Postoperative complications included endophthalmitis in one eye of one monkey and ERM in three eyes of two monkeys. For the monkey with endophthalmitis, we diagnosed this intraocular inflammation as sterile endophthalmitis because it recovered completely without any treatment (Figure 7A,B), and histological analyses showed well-preserved retinal tissue (Figure 7C). This was probably particle-induced sterile endophthalmitis, because it has been reported that IVTA includes particles that can induce sterile endophthalmitis [35]. Coincidentally, Drug-LGIR tests indicated that triamcinolone was not suppressive in this monkey (HM-8). This result suggests that we may be able to detect allergies to triamcinolone in advance with this in vitro test.

ERM was developed postoperatively in three eyes of two monkeys. Wong et al. reported that intravitreal injection of human RPE cells into rabbit eyes could cause the development of ERM [36]. Schwartz et al. reported that complications of human ESC-RPE subretinal transplantation included the development of preretinal pigmented tissues, presumably by the migration of the cells that remained in the vitreous cavity after reflux from the subretinal space or the cells that were accidentally injected into the retinal space [25], which might form ERM. The ERM tissues examined in our experiments included inflammatory cells (Appendix A). Thus, to prevent ERM formation, irrigation to clear refluxed cells after the delivery of a cell suspension into the subretinal space, together with anti-inflammatory therapies, may be effective.

Schwartz et al. transplanted human ESC-RPE cells into the subretinal space of nine patients with Stargardt’s disease and in nine patients with dry AMD [25]. Visual improvement occurred in 10 of these patients with a lower number (5~15 × 10^4^ cells) of RPE cells transplanted. In our study, we transplanted 1 × 10^6^ cells of the iPSC-RPE cells into the subretinal space, which may be a higher than normal number of transplanted cells for primates. When a larger number of cells is transplanted, it is likely that a much more severe inflammation (i.e., immune rejection) will occur in the eye postoperatively. However, since our goal was to determine the mechanisms of the RPE-related immune attacks after transplantation and the immunological properties of human iPSC-RPE cells, we intentionally used larger amounts of human RPE cells in our experiments. As intended, almost all of the transplanted monkeys had immune rejections and the grafts did not survive, which allowed us to analyze the RPE-related immunological events. Based on these results, in human clinical studies, we will need to carefully consider the appropriate number of cells to transplant, which should depend on the objective of the surgery. Accordingly, we have to consider how to use immunosuppressive drugs, which in part depends on the number of the graft cells. 

## 4. Material and Methods

### 4.1. RPE Cell Transplantation in Monkeys

Human RPE cells were differentiated from iPSC as previously described [18,19,37]. Subsequently, we transplanted the human iPSC-RPE cell suspension into the subretinal space of 8 cynomolgus monkeys (12 eyes, xenografts). After anesthetizing all of the cynomolgus monkeys with a mixture of ketamine 5 mg/kg and xylazine 1 mg/kg, their pupils were dilated with 0.5% tropicamide and 0.5% phenylephrine hydrochloride. 

For the transplantation of human iPSC-RPE cells, posterior vitreous detachment was followed by complete vitrectomy (Accurus^®^, Alcon, Fort Worth, TX, USA). After creation of a retinal bleb by subretinal injection of intraocular irrigating solution (BSS plus®, Alcon), human iPSC-RPE cells (1 × 10^6^ cells in 200 μL) were transplanted into several subretinal spaces of 12 eyes of 8 monkeys with 25-gauge/38-gauge cannula (Poly Tip^®^ Cannula 25 g/38 g, MedOne, Sarasota, FL, USA). The human iPSC-RPE cells were labeled with either PKH (Sigma-Aldrich, St. Louis, MO, USA) or eGFP, as previously described [18] for traceability.

No systemic immunosuppressive drugs were administered in any of the primates except one (Appendix A). All animal experiments were performed according to the guidelines for animal experiments of RIKEN Center for Biosystems Dynamics Research and were approved by the Animal Experiment Committee of the RIKEN Kobe Institute (Approval ID: A2008-02-15, 05/08/2019).

### 4.2. Local Steroid and Oral Cyclosporine A Administration

At the end of surgery for RPE cell transplantation, all monkeys except one (HM-4) received 2 mg (50 μL) of IVTA (MaQaid, Wakamoto, Tokyo, Japan) (Appendix A). In combination with the IVTA treatment, 3 eyes of 2 monkeys received 20 mg (500 μL) STTA (MaQaid) at 4 and 12 weeks after surgery (Appendix A). One of the monkeys (HM-5) received oral cyclosporine A for every day before and after the surgery. Cyclosporine A was mixed with food and administered daily.

### 4.3. Clinical Evaluation

The transplanted grafts were monitored using color fundus photographs (Kowa, Aichi, Japan), FA (RetCamII^®^, Clarity, Pleasanton, CA, USA), and OCT (Nidek, Aichi, Japan). After 3–8 months, monkeys were sacrificed and enucleated. After fixation, all eyes were subjected to IHC and H&E staining.

### 4.4. Histological Assessment

Enucleated eyes were fixed in formaldehyde (Super Fix, Kurabo, Osaka, Japan) for 7 days and then embedded in paraffin (Sigma-Aldrich) [20,21]. Paraffin sections were sliced into 10 µm-thick sections and prepared as a series of five sequential slides with an autoslide preparation device (Kurabo). We first performed H&E staining before IHC evaluation. Retinal sections were blocked with 5% goat serum in PBS for 1 h at room temperature. Primary antibodies to the following antigens were applied: CD3 (host: rabbit, Abcam, Cambridge, UK), Iba1 (host: rabbit, Wako, Osaka, Japan), MHC class II (MHC-II; host: mouse, DakoCytomation, Glostrup, Denmark), CD20 (host: rabbit, Abcam), immunoglobulin G (IgG; host: rabbit, Abcam), NKG2A (host: rabbit, Abcam), and IFN-γ (host: mouse, R&D Systems, Minneapolis, MN, USA). All of the antibodies were incubated at 4 °C overnight. After washing, sections were incubated with secondary antibodies for 1 h at RT and counterstained with DAPI (Life Technologies, Carlsbad, CA, USA). Images were acquired using a confocal microscope (LSM700, Zeiss, Jena, Germany) [20,21].

### 4.5. Drug-LGIR Tests and FACS

The in vitro rejection assay LGIR tests were performed as previously described [19,20,21,22] using PBMC from healthy or transplanted monkeys as host immune cells (effector lymphocytes) and human iPSC-RPE cell lines as transplanted cells (target iPSC-RPE cells). They were cocultured for 5 days, and the responses of the effector lymphocytes against the target RPE cells in vitro was evaluated by Ki-67 FACS analysis. Procedure of Ki-67 staining and antibody information were described previously [19,20,21,22]. Briefly, PBMC cocultured with iPSC-RPE cells or allogenic B cells (positive control) were harvested, then fixed and permeabilized with 70% ethanol at −20 °C for 1 h, and then labeled with phycoerythrin (PE)-labeled anti-Ki-67 (BioLegend, CA, USA) together with either APC-labeled anti-CD4, APC-labeled anti-CD8a, APC-labeled anti-CD11b, FITC-labeled anti-CD20, or APC-labeled anti-NKG2A at room temperature for 40 min. All samples were analyzed on a FACSCanto II flow cytometer (BD Bioscience, CA, USA). Data were analyzed using FlowJo software version 9.3.1 (BD Bioscience).

In the Drug-LGIR tests, dexamethasone (Sigma-Aldrich), prednisolone (Sigma-Aldrich), betamethasone (Sigma-Aldrich), hydrocortisone (Sigma-Aldrich), triamcinolone (MaQaid, Wakamoto), and cyclosporine A (Sigma-Aldrich) (all 1 μg/mL) were added to the culture medium. In preliminary experiments to optimize the drug concentration, we cultured monkey PBMC in the presence of the drugs for 5 days, then took the supernatants, and evaluated the secretion of IFN-γ by a monkey IFN-γ ELISA kit (R&D systems) following the manufacturer’s instruction.

## 5. Conclusions

We transplanted human iPSC-RPE cells into the subretinal space of cynomolgus monkeys. All eyes of the monkeys exhibited postoperative inflammation, although the degree of inflammation varied among individuals. Local steroid therapy, as well as systemic immunosuppressive drugs, was effective in reducing postoperative inflammation and maintaining less or no inflammation, which led to longer graft survival. The appropriate dose and administration period of steroids and immunosuppressive drugs still need to be experimentally examined. In this study, we established a novel examination test for personalized treatment of retinal patients that suffer from immune attacks after iPSC-RPE cell transplantation. Immune attacks in the eye hamper the survival of the grafts. Once the best medication for each patient is experimentally identified by this in vitro test, we should use that medication to control the RPE-related immune rejections for that patient. 

## Figures and Tables

**Figure 1 ijms-21-03077-f001:**
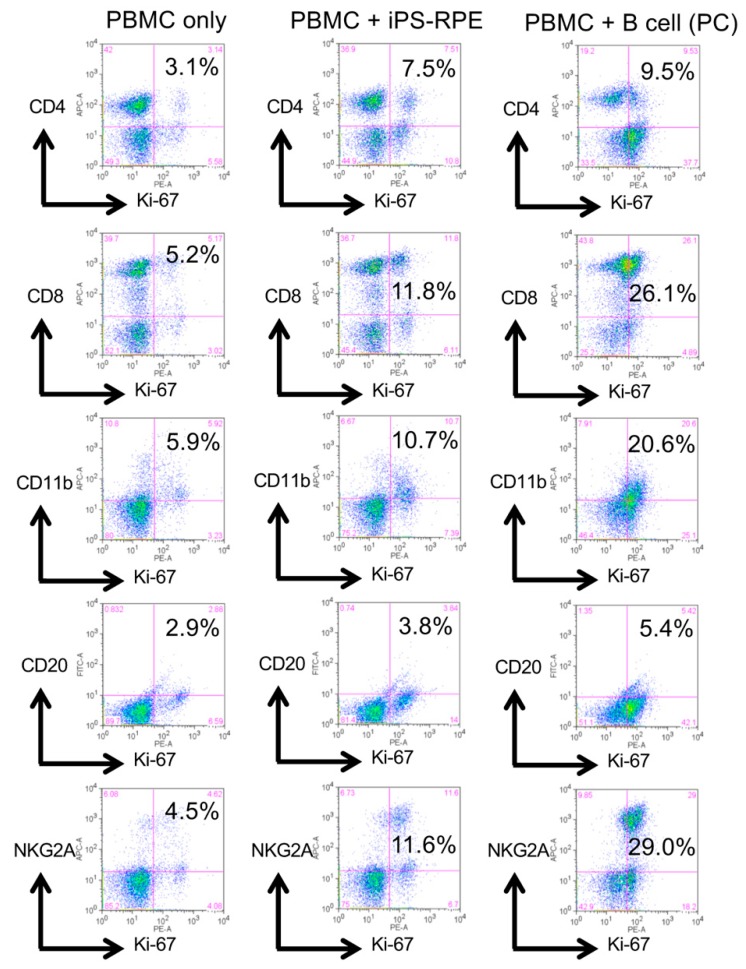
Representative results of LGIR with peripheral blood mononuclear cells (PBMC) from a monkey. PBMCs (2 × 10^6^ cells/well) from a healthy monkey HM-1 were cultured with iPSC-RPE cells for 5 days. Before the assay, iPSC-RPE cells were irradiated (20 Gy) and 1 × 10^4^ cells were used for a 24-well culture plate. After 5 days of coculture, PBMC were harvested and stained with anti-CD4 (helper T cell-marker), anti-CD8 (cytotoxic T cell-marker), anti-CD11b (monocyte-, macrophage-, NK cell-, and granulocyte-marker), anti-CD20 (B cell-marker), anti-NKG2A (natural killer (NK) group 2 member A; NK cell-marker), and anti-Ki-67 (proliferation marker) antibodies. As a positive control (PC), irradiated allogenic B cells were used. The samples were analyzed by a fluorescence-activated cell sorting (FACS) flow cytometer. Numbers (%) in the scatterplots indicate double-positive cells (e.g., CD4/Ki-67).

**Figure 2 ijms-21-03077-f002:**
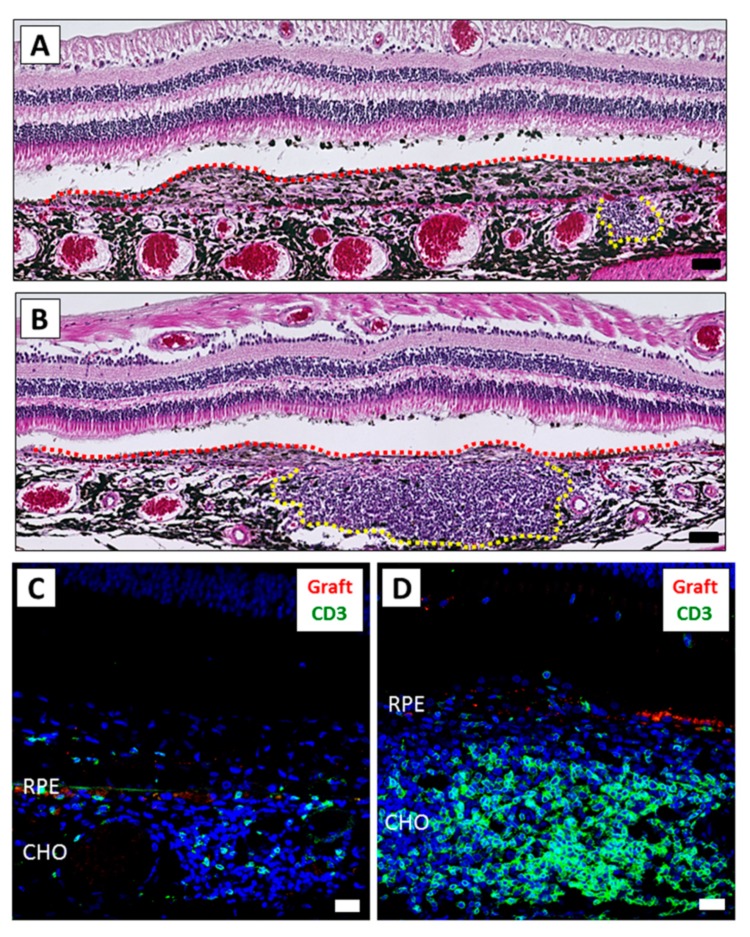
The inflammatory difference between the right and left eyes after iPSC-RPE transplantation. The eyes of monkey HM-1 were histologically examined. (**A**,**B**) H&E staining of the right (**A**) and left (**B**) eye. Graft cells are indicated by the red dot-line. The mass of inflammatory cells is indicated by the yellow dot-line. The left eye exhibited severe inflammation compared to the right eye. Scale bars: 50 μm. (**C**,**D**) IHC of the right (**C**) and left (**D**) eye. A larger amount of CD3^+^ infiltration (green) was detected in the left eye compared to the right eye. PKH-positive live grafted RPE cells (red) were not detected. Scale bars: 20 μm. CHO: choroid.

**Figure 3 ijms-21-03077-f003:**
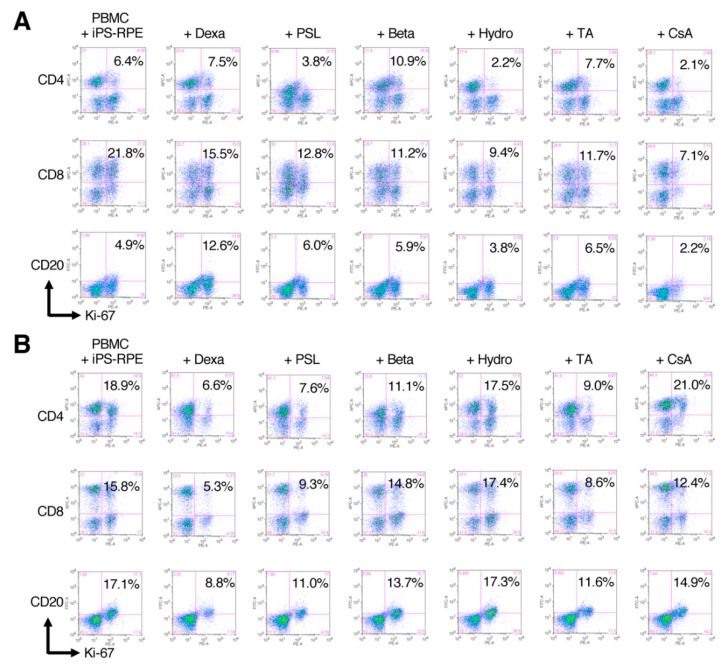
Representative FACS results of Drug-LGIR in PBMC from healthy or transplanted monkeys. In the Drug-LGIR assay, PBMCs were cultured with human iPSC-RPE cells in the presence of indicated drugs for 5 days, and the suppressive effects of the drugs were evaluated by FACS of Ki-67 expressing cells. (**A**) The results of monkey HM-5 before transplantation. Hydrocortisone, prednisolone, and cyclosporine A exhibited significant suppression of the proliferation of the immune cell. By contrast, dexamethasone, betamethasone, and triamcinolone were not suppressive. (**B**) The results of monkey HM-2 that showed immune rejection against human iPSC-RPE cells in vivo. In the results of the in vitro assay, dexamethasone exhibited significant suppression of the immune reaction. Prednisolone and triamcinolone also exhibited suppressive effects. However, other drugs were not suppressive. Dexa: dexamethasone, PSL: prednisolone, Beta: betamethasone, Hydro: hydrocortisone, TA: triamcinolone, and CsA: cyclosporine A.

**Figure 4 ijms-21-03077-f004:**
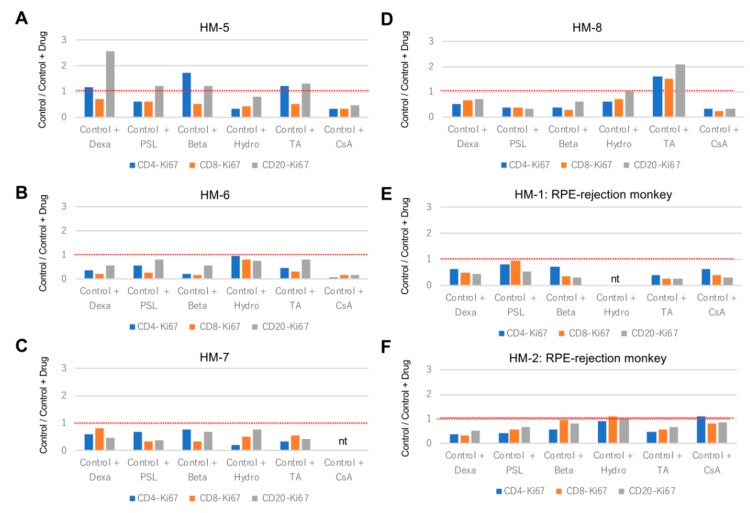
Summary of Drug-LGIR assay in monkeys. (**A**–**F**) The results of Drug-LGIR assay with dexamethasone (Dexa), prednisolone (PSL), betamethasone (Beta), hydrocortisone (Hydro), triamcinolone (TA), and cyclosporine A (CsA) in PBMC from six monkeys: HM-5 (**A**), HM-6 (**B**), HM-7 (**C**), and HM-8 (**D**) that were not transplanted (*n* = 4) and HM-1 (**E**) and HM-2 (**F**) that showed RPE-rejection after transplantation (*n* = 2). The rate of proliferative immune cells indicated by expression of Ki-67 by FACS are shown in bar graphs to evaluate the suppressive effect of each drug. Red dotted-lines indicate the baseline (Control = 1.0, PBMC + iPS-RPE cells without drug). nt: not tested.

**Figure 5 ijms-21-03077-f005:**
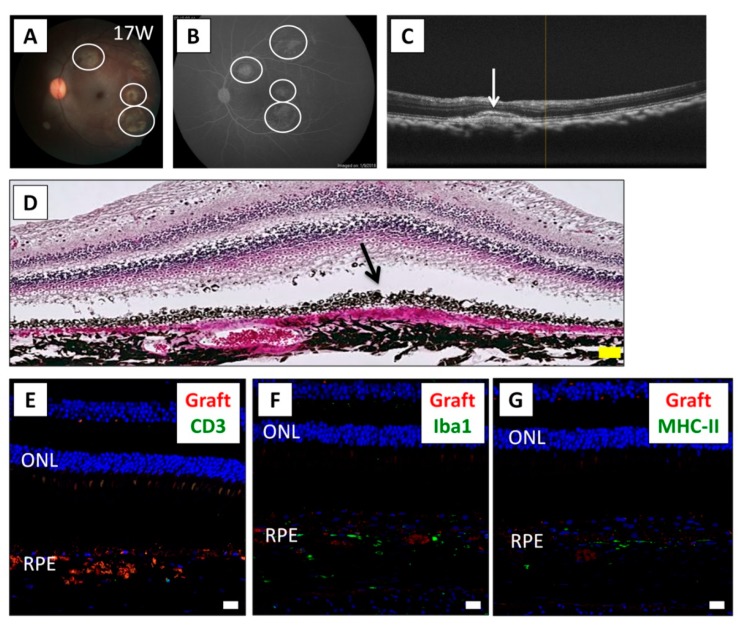
Results of iPSC-RPE transplantation while using local steroid therapy. (**A**) The fundus photograph of monkey HM-6 at 17 weeks after surgery, who was subjected to human iPSC-RPE transplantation with IVTA injection on day 0 and STTA injections at 4 and 12 weeks after transplantation. White circles show the transplanted sites. (**B**) Fluorescein angiography (FA) at late phase showed no hyperfluorescence around the grafts. No FA leakage was observed during the evaluation period. (**C**) Optical coherence tomography (OCT) revealed the presence of the cell aggregates of graft cells (arrow) in the subretinal space. (**D**) In H&E staining at 17 weeks after surgery, the graft cells (arrow) were observed in the subretinal space. Inflammatory cells were not obviously recognized. The overlying neurosensory retina was well preserved. Scale bar: 50 μm. (**E**–**G**) IHC evaluation at 17 weeks after surgery. T cells were not observed in the retina (**E**). Although Iba1^+^ (**F**) and MHC class II^+^ cells (**G**) were observed around the grafts, little invasion of these antigen-presenting cells was observed throughout the retinal sections. PKH^+^ RPE cells (red), indicating live grafts, were detected in the subretinal space. Nuclei were stained by DAPI (blue). Scale bars: 20 μm. ONL: outer nuclear layer.

**Figure 6 ijms-21-03077-f006:**
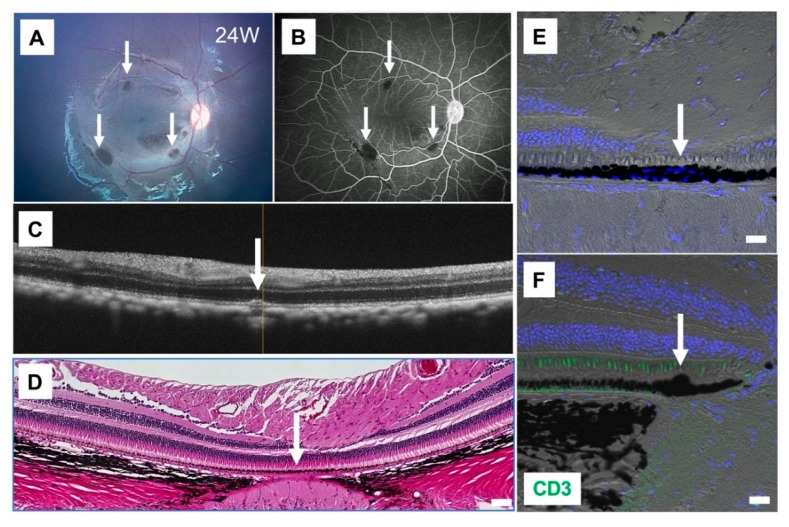
Results of iPSC-RPE transplantation while using cyclosporine A. (**A**) The fundus photograph of monkey HM-5 at 24 weeks after surgery, who was subjected to human iPSC-RPE transplantation with administration of cyclosporine A before the transplantation. Arrows show the transplanted sites. (**B**) Fluorescein angiography (FA) at late phase showed no hyperfluorescence around the grafts (arrows). No FA leakage was observed during the 24-week evaluation period. (**C**) Optical coherence tomography (OCT) showed the presence of cell aggregates of graft cells (arrow) in the subretinal space. (**D**) The graft cells (arrow) but not inflammatory cells were observed in the subretinal space by H&E staining. Scale bar: 50 μm. (**E**–**F**) IHC with CD3 (green) and DAPI (blue). RPE grafts were detected in the subretinal space (arrows). CD3^+^ T cells were not observed in the retina (**F**). Scale bars: 20 μm.

**Figure 7 ijms-21-03077-f007:**
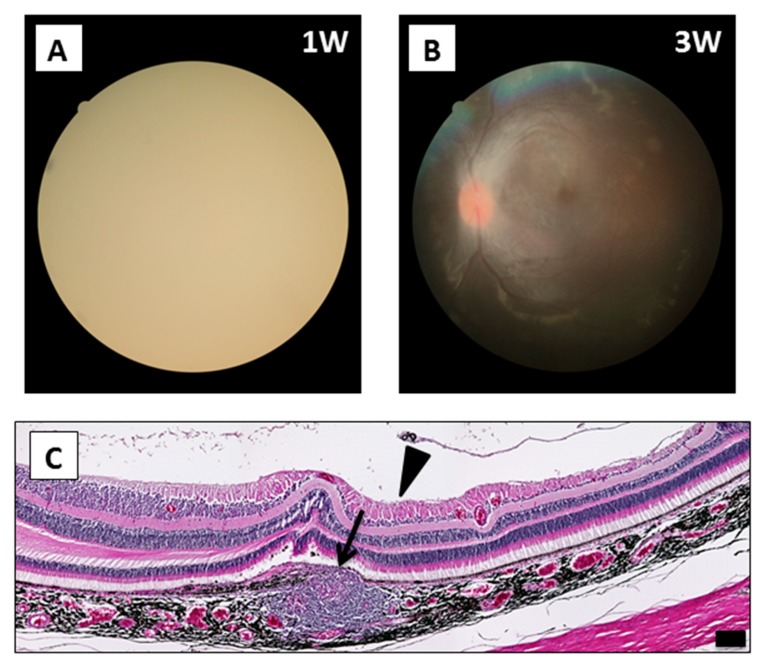
Surgical complications seen after the iPSC-RPE transplantation: sterile endophthalmitis. (**A**) The fundus photograph of monkey HM-8 at 1 week after transplantation. Endophthalmitis was developed, which made the fundus invisible. (**B**) The inflammation disappeared around 3 weeks after surgery without any treatment. (**C**) H&E staining at 12 weeks after transplantation showed the graft (arrow) and epiretinal membrane (ERM; arrowhead). We diagnosed this case as triamcinolone-related sterile endophthalmitis. Scale bar: 100 μm.

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
