# Peer review of "A Strategy for Personalized Treatment of iPS-Retinal Immune Rejections Assessed in Cynomolgus Monkey Models"

_ijms, 2020, doi:10.3390/ijms21093077_

Round 1
Reviewer 1 Report
This current manuscript by Fuji et al describes novel technique enabling to evaluate therapeutic potential of commonly used immunosuppressant drugs along with the transplants prior to the treatment, using host immunocompetent cells and transplant cells. As different patients respond to the specific treatment differently, this method could allow to control the immune attack after transplantation in personalized way. The outcome of this study performed on eight monkeys are very promising. Although this study is very interesting and informative, the manuscript would benefit from multiple clarifications listed below:
- The authors use many acronyms in this manuscript, which are potentially clear for researchers in immunology field but not necessary to the broaden audience. I would suggest to add full names to the following acronyms: MHC, PBMC, OCT, H&E, IHC, PKH, NKG2A, NK, IFN-g, TNF-a, IL-2 when they appear for the first time in the text.
- Along the same line, CHO in the Figure 2, ONL in the Figure 5, FA in the legend of the Figure 6.
- The legend of the Figure 3 lacks explanation of the abbreviations for tested drugs. However, these can be found in the Figure 4. Please add them to also to the Figure 3.
- Red circles in panel A and B of the Figure 5 should be replaced with white color. Red color is not very well visible in this dark background image.
- Evaluation of drug effects needs more explanation. Did the author tested a range of concentrations for each drug or only one concentration as stated in the methods section? If one concentration then why this particular one? This is rather uncommon to just test single drug concentration to evaluate its effectiveness. Therefore, an explanation is necessary. This explanation should appear in the results section. Was drug toxicity evaluated?
- On the page 8 it is stated that cyclosporine A was administered before the transplantation but when exactly, how many hours, days… and what was the rout of administration. Was it only single administration? In general, more details for drug administration should be added to this manuscript.
- In the methods section it is stated that monkeys were anesthetized with a mixture of ketamine and xylazine but what was the concentration used?
- Because this manuscript is describing novel Drug-LGIR methodology it would tremendously benefit if more details would be included on the LGIR technique instead of sending the reader to previous publications. More details to drug evolution would be valuable as well. For example, after addition of the drug to cell culture when its effect was evaluated, how long cells were incubated with the drug. So how long patient would have to wait for the drug prescreening before the transplant.
- In general, the Material and Methods section is missing description of OCT and SLO imaging, histological assessment and FACS experiment.
- All paragraphs numbering is wrong. Please carefully correct them.
- On the page 5 line 161 "…the immune cells as assed by Ki-67…" should be assessed
Author Response
We would like to thank the reviewers for their careful review of our manuscript and their helpful comments. We have revised our manuscript in line with the reviewers’ suggestions.
The authors use many acronyms in this manuscript, which are potentially clear for researchers in immunology field but not necessary to the broaden audience. I would suggest to add full names to the following acronyms: MHC, PBMC, OCT, H&E, IHC, PKH, NKG2A, NK, IFN-g, TNF-a, IL-2 when they appear for the first time in the text.
Response: Thank you for your comments. We described the full names of the following acronyms: MHC, PBMC, OCT, H&E, IHC, NKG2A, NK, IFN-g, TNF-a, IL-2, FA, and others in the revised manuscript.
PKH is a product name of a dye for fluorescent labeling of live cells provided by Sigma (we asked the vendor and found that PKH was the full name). We added a sentence in the result section of the revised manuscript to clarify that it is a commercially available dye (page 4): iPSC-RPE cells were pre-labeled with PKH-fluorescent dye that allows tracking of live cells.
Along the same line, CHO in the Figure 2, ONL in the Figure 5, FA in the legend of the Figure 6.
Response: We also added the full names of these in the revised manuscript.
The legend of the Figure 3 lacks explanation of the abbreviations for tested drugs. However, these can be found in the Figure 4. Please add them to also to the Figure 3.
Response: Thank you for your comments. We added explanation of the abbreviations in the legend of Figure 3.
Red circles in panel A and B of the Figure 5 should be replaced with white color. Red color is not very well visible in this dark background image.
Response: As per your suggestion, we have changed Figure 5 with white color circles.
Evaluation of drug effects needs more explanation. Did the author tested a range of concentrations for each drug or only one concentration as stated in the methods section? If one concentration then why this particular one? This is rather uncommon to just test single drug concentration to evaluate its effectiveness. Therefore, an explanation is necessary. This explanation should appear in the results section. Was drug toxicity evaluated?
Response: Thank you for the comments. We previously tested a range of concentrations for each drug before developing the Drug-LGIR assay. Although we did not test the drug toxicity, we evaluated the best concentration of each drug for the in vitro assay. Please see new supplemental Fig. 5 (Fig. S5) and text, page 6.
On the page 8 it is stated that cyclosporine A was administered before the transplantation but when exactly, how many hours, days… and what was the rout of administration. Was it only single administration? In general, more details for drug administration should be added to this manuscript.
Response: As per your suggestion, we added the information in the result section (page 9): We administrated cyclosporine A before & after the transplantation daily by oral administration with food mixture.
In the methods section it is stated that monkeys were anesthetized with a mixture of ketamine and xylazine but what was the concentration used?
Response: As per your suggestion, we added the information in the method section (page 14): a mixture of ketamine 5mg/kg and xylazine 1mg/kg.
Because this manuscript is describing novel Drug-LGIR methodology it would tremendously benefit if more details would be included on the LGIR technique instead of sending the reader to previous publications. More details to drug evolution would be valuable as well. For example, after addition of the drug to cell culture when its effect was evaluated, how long cells were incubated with the drug. So how long patient would have to wait for the drug prescreening before the transplant.
Response: Thank you for your comments. The drugs are added to the culture for 5 days and the following assay takes only one day, so we can obtain the data in a week (around 6 days). We described this in the results section (page 2-3) and also added a sentence in the discussion section (page 12) as follows: Since the drug in vitro test is for 5 days followed by one-day experiment of FACS, it takes only a week to obtain the result of the drug prescreening before transplantation.
In general, the Material and Methods section is missing description of OCT and SLO imaging, histological assessment and FACS experiment.
Response: As per your suggestion, we added more detailed information in the method section (page 14-15).
All paragraphs numbering is wrong. Please carefully correct them.
On the page 5 line 161 "…the immune cells as assed by Ki-67…" should be assessed
Response: Thank you for pointing out. These are corrected in the revised manuscript. Thank you.
Reviewer 2 Report
In the present manuscript the authors present data in non.human primate models with clinical relevance and good translational data for the field of RPE cell transplantation for retinal degenerations. The data and conclusions presented are in agreement.
Some minor comments are still to be addressed:
- The first paragraph of the introduction is stated as an opinion, without references. This is not appropriate and must be re-written/re-formatted.
- in line 75, regarding immunorejection in rabbit models of RPE transplantation, the authors used hight selective referencing and ignored works such as DOI: 10.1167/iovs.16-20738 and DOI: 10.1016/j.stemcr.2020.02.006, among others
- direct attention to accronyms. All must be dysambiguated at the first entry in text. Examples include: MHC, line 25; PBMC, line 94; FA, line 114; PKH, line 124...
- in line 115, the OCT data not shown is of relevance for the study and should be added as supplementary information.
- Fig 3, legend and results text (lines 160-163) are in contradiction to each other. Please correct.
- Figure legend of fig 3 is incomplete with regards to treatments/drugs. refer to fig legend 4 and disclose acronyms in all fig legends for readability.
Author Response
Point-by-point responses to reviewers: Manuscript ID: ijms-787087. R1
We would like to thank the reviewers for their careful review of our manuscript and their helpful comments. We have revised our manuscript in line with the reviewers’ suggestions.
The first paragraph of the introduction is stated as an opinion, without references. This is not appropriate and must be re-written/re-formatted.
Response: Thank you for your comments. As suggested, we added several references in the introduction of the revised manuscript (page 2).
in line 75, regarding immunorejection in rabbit models of RPE transplantation, the authors used hight selective referencing and ignored works such as DOI: 10.1167/iovs.16-20738 and DOI: 10.1016/j.stemcr.2020.02.006, among others
Response: As per your suggestion, we added the two references in the introduction of the revised manuscript (page 2).
direct attention to accronyms. All must be dysambiguated at the first entry in text. Examples include: MHC, line 25; PBMC, line 94; FA, line 114; PKH, line 124...
Response: We described the full names of the following acronyms in the revised manuscript: MHC, PBMC, OCT, H&E, IHC, NKG2A, NK, IFN-g, TNF-a, IL-2, FA, and others. PKH is a product name of a dye for fluorescent labeling of live cells provided by Sigma (we asked the vendor and found that PKH was the full name). We added a sentence in the result section of the revised manuscript to clarify that it is a commercially available dye (page 4): iPSC-RPE cells were pre-labeled with PKH-fluorescent dye that allows tracking of live cells.
in line 115, the OCT data not shown is of relevance for the study and should be added as supplementary information.
Response: Thank you for your comments. We added the data of OCT and FA in new supplemental Fig. 2 (Fig. S2).
Fig 3, legend and results text (lines 160-163) are in contradiction to each other. Please correct.
Response: Thank you for your comments. Fig. 3 legend was miswritten. We have revised the legend.
Figure legend of fig 3 is incomplete with regards to treatments/drugs. refer to fig legend 4 and disclose acronyms in all fig legends for readability.
Response: Thank you for your advice. We have revised the figure legend of Fig. 3. Also, we described all acronyms in all fig legends. Thank you.